# Information, involvement, self-care and support—The needs of caregivers of people with stroke: A grounded theory approach

Elton H. Lobo[1,2]*, Anne Frølich[2,3], Mohamed Abdelrazek[1], Lene J. Rasmussen[4,5], John Grundy[6], Patricia M. Livingston[7], Sheikh Mohammed Shariful Islam[8], Finn Kensing[9]

**1** School of Information Technology, Deakin University, Geelong, VIC, Australia, **2** Department of Public Health, University of Copenhagen, Copenhagen, Denmark, **3** Innovation and Research Centre for Multimorbidity, Slagelse Hospital, Region Zealand, Denmark, **4** Department of Cellular and Molecular Medicine, University of Copenhagen, Copenhagen, Denmark, **5** Center for Healthy Aging, University of Copenhagen, Copenhagen, Denmark, **6** Faculty of Information Technology, Monash University, Melbourne, VIC, Australia, **7** Faculty of Health, Deakin University, Geelong, VIC, Australia, **8** Institute for Physical Activity and Nutrition (IPAN), Deakin University, Geelong, Australia, **9** Department of Computer Science, University of Copenhagen, Copenhagen, Denmark

\* elobo@deakin.edu.au

**Data Availability Statement:** All relevant data are within the manuscript and its Supporting Information files.

## Abstract

### Background

Globally, stroke is a leading cause of death and disability, with most care undertaken by caregivers who are generally family and friends without prior experience of care. The lack of experience or unpreparedness results in feelings of uncertainty, burnout, anxiety, burden, etc. Hence, it is necessary to identify the needs of caregivers to better support them in their caregiving journey and improve the quality of care delivered.

### Methods

The study employed a grounded theory methodology that utilizes information gathered from literature reviews and social media to represent the needs and create a storyline visually. The storyline is further refined and evaluated using an online survey of 72 participants recruited through online stroke caregiving communities.

### Results

The study identified four core categories of needs: (i) Information: sufficient information delivered in layman's terms based on the individual situation of the caregiver and survivor through oral and hands-on demonstrations, (ii) Involvement: inclusion in the decision-making processes at different stages of recovery through face-to-face communication at the hospital, (iii) Self-care: ability to engage in work and leisure activities, (iv) Support: receive support in the form of resources, services and finances from different other stakeholders.

**Funding:** This study was supported through doctoral scholarships from the School of Information Technology, Deakin University, and the Department of Public Health, University of Copenhagen. Further, Prof John Grundy is supported by ARC Laureate Fellowship FL190100035 and Dr Islam is supported by NHMRC Emerging Leadership Fellowship.

**Competing interests:** The authors have declared that no competing interests exist.

## Conclusions

There is a need to create a caregiver-centered approach in stroke recovery to ensure limited obstruction to care and reduced uncertainty in stroke recovery. Moreover, through the inclusion of caregivers in stroke recovery, it may be possible to reduce the burden of care to the caregiver and ensure the satisfaction of the healthcare system throughout stroke recovery.

## Introduction

Globally, there are over 13.7 million new incident cases of stroke each year, with more than 116 million years of healthy life is lost due to stroke-related deaths and disabilities [1]. With the recent advancements in the medical field, the stroke mortality rate has decreased [2], resulting in more than 80 million people currently living after experiencing a stroke [1]. Of these survivors, up to 50% of individuals are chronically disabled [3], with family members and/or friends assuming the role of caregiver to provide care [4].

Family caregivers play a central role in post-stroke care [5]. However, caregiving is a complex concept and is dependent on the condition of the survivor (i.e., age, impairments and living situations), individual characteristics (i.e. personal beliefs, coping styles and social expectations) and the relationship between the survivor and the caregiver [6, 7]. Moreover, since the occurrence of stroke is sudden [6], caregivers often have to adjust to the diagnosis [8], assume new roles and responsibilities [9] and face the challenge of becoming a caregiver with limited or no preparation [10]. This leads to a substantial increase in caregiver strain or burden, leaving the caregiver to feel abandoned and unsupported [11].

As a result, several researchers have attempted to explore the needs of stroke caregivers at different stages of stroke. For example, a systematic review by Luker et al. [12] highlighted the needs for stroke caregivers during in-patient rehabilitation. At the same time, a longitudinal study by Tsai et al. [13] described the changing needs of stroke caregivers at different stages of stroke recovery.

Although numerous methodologies, in the past, have been implemented that understand the caregiver's needs and experiences during stroke recovery, an updated understanding that is grounded in data is necessary to create effective interventions to support the caregiver during the transition into the caregiver role. Hence, this study utilizes a grounded theory methodology that combines data acquired from literature and social media sources to identify needs specific to the caregiver and refine and evaluate these needs using an online-based survey to ensure comprehensiveness.

## Materials and methods

### Study design

The study was designed and conducted based on the grounded theory methodology [14], which is a well-known methodology that sets out to discover or construct theory from systematically obtained data and analyzed using comparative analysis techniques [15]. Grounded theory can utilize both qualitative and quantitative data generated from different sources such as interviews, questionnaires, grey literature, surveys, memos, blogs and so on [15] to investigate a particular phenomenon in diverse environments to develop an explanatory theory [16].

## Data collection

Theoretical sampling [17] was used to follow leads in data collected based on concepts developed from an initial data analysis. This method considers following data to expand and refine the existing theories during analysis [17]. The authors utilized theoretical sampling in three phases conducted between September 2020 to May 2021 starting with a literature review (Phase 1) to identify the target user groups and relevant concepts. Initial data collected was expanded and used to extract critical data from social media (Phase 2) and caregiver surveys (Phase 3). Phases 1 and 2 were completed in English, while phase 3 was conducted in English and Danish. Data collected in Danish was translated into English by the second author (AF).

Five authors (EL, MA, FK, AF and SI) coded the data collected in English based on three essential coding schemes; initial, immediate, and advanced coding as shown in **Fig 1** to ensure the final themes identified were grounded in data. To ensure anonymity and confidentiality, potentially identifiable information related to the participants was excluded from the study.

**Phase 1: Literature review.** The first phase involves a traditional review of the literature to determine the needs and inform the type of caregiver groups to be involved in the study.

- **Selection Criteria:** Qualitative studies that included the needs, experiences, and perspectives of caregivers supporting people living with stroke at their home was eligible for inclusion. Articles were excluded if they (i) were not available in English, (ii) were protocols or abstracts, (iii) solely reported quantitative data, and (iv) reported needs, experiences, and perspectives within the hospital. Studies were excluded if they did not acquire data directly from the caregiver. Furthermore, reference lists of included reviews were screened to identify additional relevant studies.

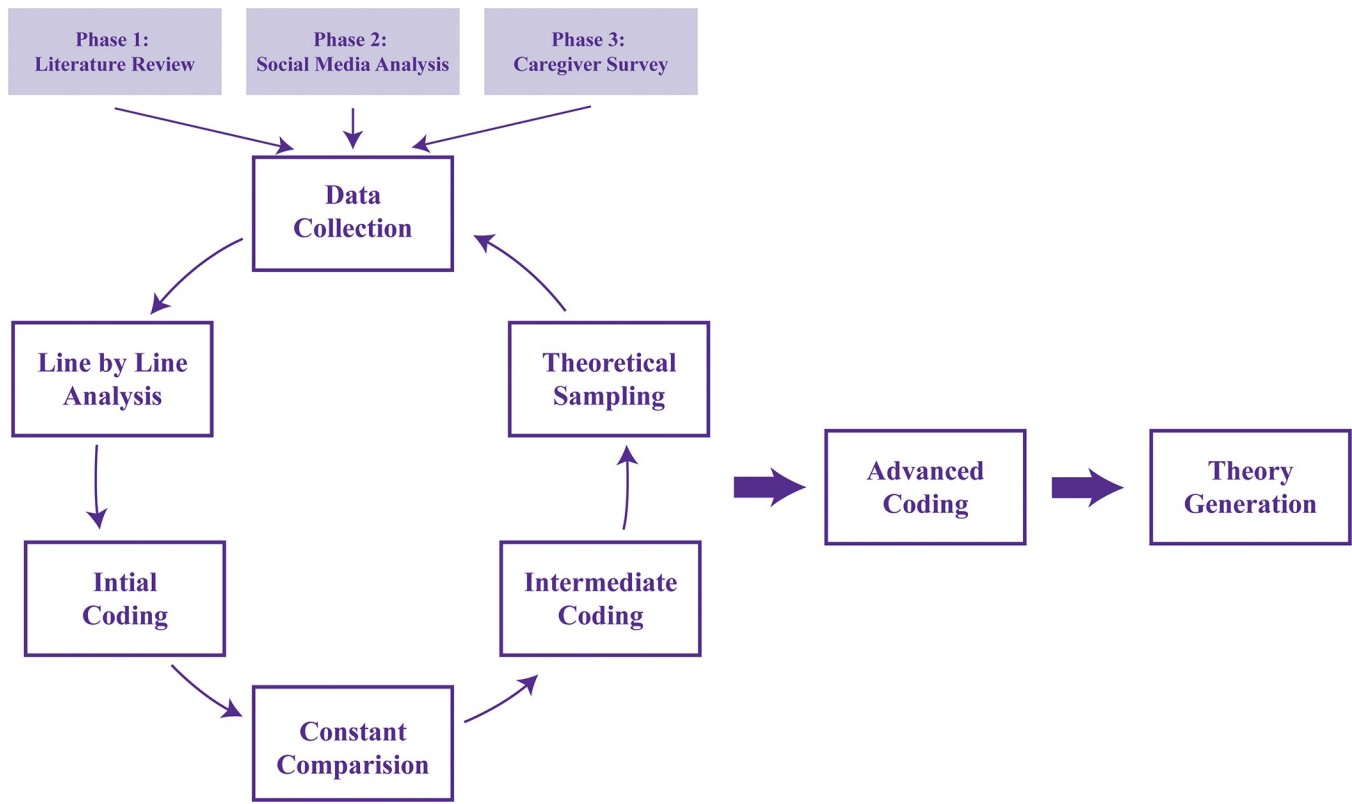

**Fig 1. Grounded theory approach to identify caregiver needs.**

- **Data Sources and Searches:** A manual search was conducted on five electronic databases: Medline, Embase, CINAHL, PsycINFO and Web of Science, from inception to September 2020 for keywords related to stroke caregiving needs, experiences and perspectives (**Table 1**) combined using AND and OR boolean operators. After removing duplicates, the primary author (EL) reviewed titles and abstracts and was supervised by another author (MA). The potentially relevant articles were downloaded in full-text and independently reviewed by two authors (EL and FK). All authors discussed any discrepancies until a consensus was achieved. **Fig 2** presents the filtration process of the review.

- **Data Extraction and Representation:** User needs data was extracted from thirty-one accepted articles by the primary author and was subsequently reviewed by another author for accuracy. The data extracted were represented using a concept mapping approach to visually illustrate different concepts into meaningful connections [18].

  **Phase 2: Social media analysis.** The data extracted from the first phase led to further data gathering. Phase two data collection was based on user posts on popular information-based social media platforms (i.e., Facebook and Twitter). These social media platforms were selected as it has in the past been used by organizations and individuals to actively engage and participate in health-care. Moreover, it can promote better health management and decision-making processes [19].

  As the data analysis proceeded, theoretical sampling was implemented to decide how to collect data based on emerging theories and categories. The data collected from social media posts was then analyzed with the previous data to create an initial storyline.

- **Selection Criteria:** Social media posts were included if they (i) were available in English, (ii) were made publicly available, (iii) included stroke caregiver discussions, and (iv) discussed stroke caregiver needs during care. The posts were excluded if they (i) included discussions from patients, clinicians, or community administrators, (ii) were not available publicly, (iii) discussed promotions of products or services, and (iv) were not related to the needs of stroke caregivers.

- **Data Sources and Searches:** A search was conducted on two social media platforms (i.e., Facebook and Twitter) from inception to January 2021 to identify relevant stroke communities using keywords from popular internet searches from December 2010 to January 2021 (**Table 2**).

**Table 1. Search terms.**

| | |
|---|---|
| **Disease** | "cerebrovascular disorders" OR "basal ganglia cerebrovascular disease" OR "brain ischemia" OR "carotid artery diseases" OR "intracranial arterial diseases" OR "intracranial embolism and thrombosis" OR "intracranial haemorrhage*" OR "stroke" OR "brain infarction" OR "cerebrovascular accident" OR vasospasm OR "vertebral artery dissection" |
| **AND** | |
| **Concept** | "transitioning from hospital to home" OR "transition home" OR "discharged home" OR outpatient OR transition |
| **AND** | |
| **Methodology** | "Qualitative Research" OR "Cohort Studies" OR "Observational Study" OR "Focus Groups" OR semi-structured OR semistructured OR unstructured OR informal OR in-depth OR indepth OR face-to-face OR structure OR guide OR interview* OR discussion* OR question?aire* |
| **AND** | |
| **Study Focus** | "Patient centered" OR patient centred OR Patient-centered OR patient-centred OR "patient satisfaction" OR "consumer satisfaction" OR "caregiver focus" OR "carer focused" OR "caregiver centred" OR "caregiver centered" OR "caregiver satisfaction" OR "carer support" OR "carer centered" OR "carer centred" OR "carer satisfaction" |

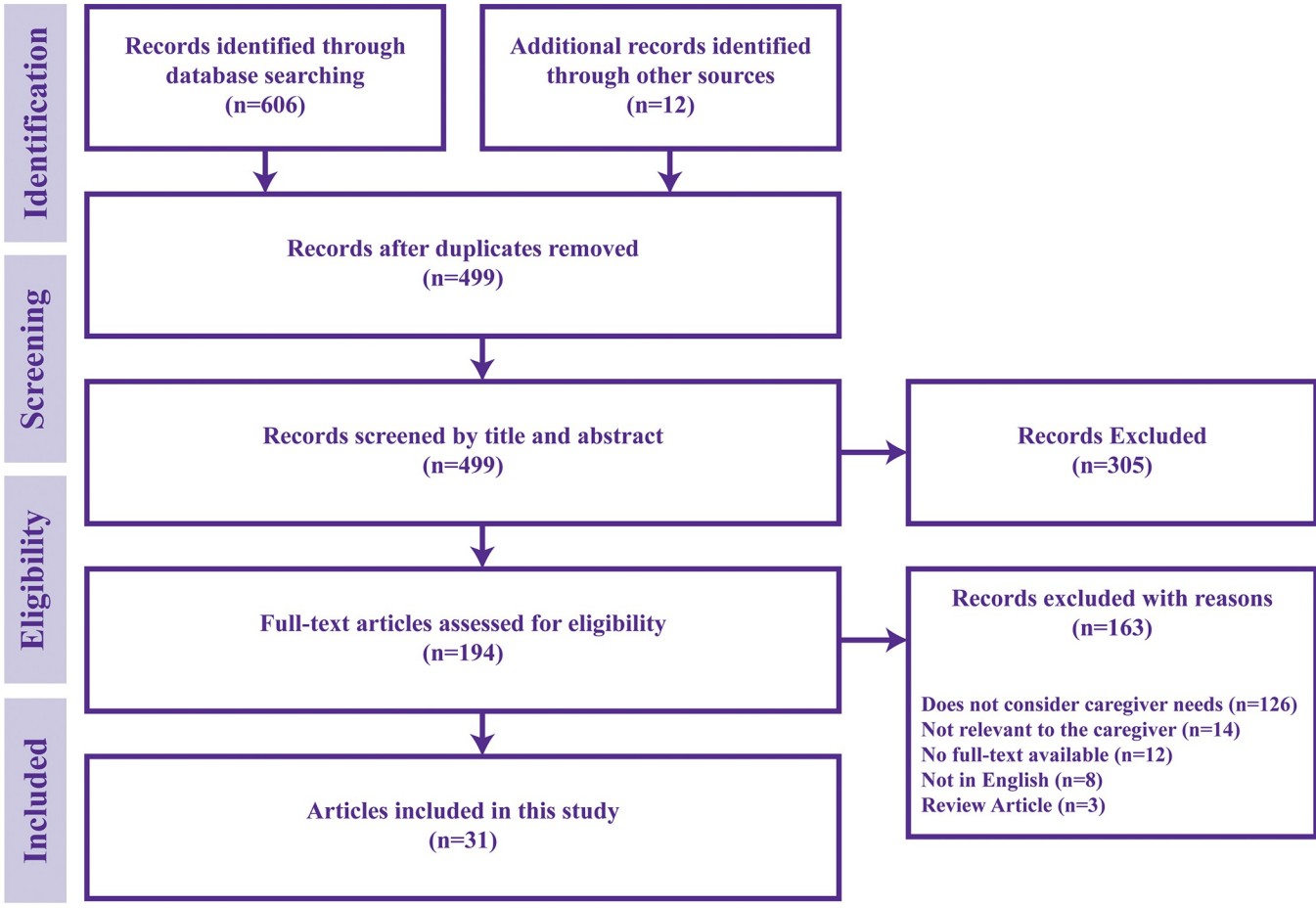

**Fig 2. Article filtration process.**

- **Data Extraction and Representation:** The primary author (EL) developed and used a Python-based web-scrapping tool that visits the identified stroke community pages, extracts user-generated posts and stores the post on a local database. These posts were filtered by the primary author (EL) based on the selection criteria and were reviewed by two author (AF and SI). The 1017 accepted posts were analyzed with the data from the previous phase to develop the initial storyline.

    **Phase 3: Caregiver survey.** In phase three, theoretical sampling is utilized to produce a theory that is grounded in data and has explanatory relationships between the theories and categories [15]. This process utilizes data collected through an anonymous online survey.

**Table 2. Keywords used in social media searches.**

| Keywords | |
| --- | --- |
| • Stroke | • Stroke Unit |
| • Stroke Care | • Traumatic Brain Injury |
| • Stroke Recovery | • Lacunar Infarct |
| • Apoplexy | • Stroke Medication |
| • Cerebrovascular Accident | • Aphasia |

- **Ethics:** Prior to phase three data collection, the study received approval from the Deakin University Human Research Ethics Committee (HREC): 2020–225.

- **Selection Criteria:** Adult caregivers of people with stroke receiving active or palliative treatment over the past five years. The caregivers were excluded is they were aged below 18 years and did not understand English or Danish language(s).

- **Participant Recruitment and Study Setting:** Caregivers were recruited through stroke health organizations, caregiver organizations and social media communities from March 2021 to May 2021 using a recruitment flyer. Interested participants were provided with a plain language statement that informed them about the research. Participants willing to engage in the survey provided digital informed consent prior to completing the anonymous online survey on Qualtrics XM.

- **Participant Demographics:** Seventy-two caregivers were recruited, including 58 female and 14 males between the ages of 24 to 83 years. The caregivers predominately cared for their partners, who were mostly male aged between 40 to 82 years. Other people with stroke cared for by the caregivers include parents, grandparents and child, within the ages of 24 to 80 years. Table 3 includes the demographics of caregivers included in this study.

- **Data Representation:** The participants received open-ended online survey questions based on the concept map and the storyline identified in the previous two phases to determine its relevance to the stroke caregiving needs. The survey included questions related to the

**Table 3. Participant demographics.**

| Caregiver | | |
|---|---|---|
| Gender | Male | 14 (19.4%) |
| | Female | 58 (80.6%) |
| Age | 18–44 | 16 (22.2%) |
| | 45–64 | 37 (51.4%) |
| | ≥65 | 19 (26.4%) |
| Living Location | Metropolitan | 36 (50.0%) |
| | Smaller City | 11 (15.3%) |
| | Rural Area | 25 (34.7%) |
| Education | Primary School | 1 (1.4%) |
| | Secondary School | 10 (13.9%) |
| | Diploma | 5 (6.9%) |
| | Bachelor's | 30 (41.7%) |
| | Master's | 14 (19.4%) |
| | Doctoral | 2 (2.8%) |
| | Other | 10 (13.9%) |
| Person being cared for | Grandparent | 2 (2.8%) |
| | Parent | 16 (22.2%) |
| | Partner | 50 (69.4%) |
| | Child | 4 (5.6%) |
| **Survivor** | | |
| Gender | Male | 51 (70.8%) |
| | Female | 21 (29.2%) |
| Age | 18–44 | 3 (4.2%) |
| | 45–64 | 33 (45.8%) |
| | ≥65 | 36 (50.0%) |

caregiver demographics (age, gender, primary language of communication, location of residence, level of education, current employment and health status), survivor demographics (age, gender, relationship to the caregiver, other comorbidities and health status), time spent caring for the survivor, time spent on other activities, understanding of stroke care processes and recovery, needs in providing care and expectations for the future.

### Data analysis

The data extracted in each phase was analyzed based on a constant comparative analysis [20] involving three prominent coding processes, i.e., initial, intermediate and advanced coding as illustrated in **Fig 1**. All data collected was organized and managed using NVivo 12. Based on the constant comparative method, the "core category" of the grounded theory is identified from different parts of the data, including emerging codes, properties, dimensions and categories as well as some parts of the data which are compared with the identified themes to identify variations, differences and similarities [20]. This means that the incoming data or new codes in each phase were compared with those from either the same or previous phase(s) to generate a theory.

The raw data is broken down into fragments and labeled to form new codes in the initial coding process. The process of labeling stems from conceptual actions or observable needs. The data from the initial coding is condensed to create a more focused coding. At this stage, the codes identified formed significant categories. The categories undergo constant comparative analysis until the advanced coding phase. In the advanced coding phase, the high-level codes look to address the theory, i.e., the needs of stroke caregivers during care.

## Results

Findings highlight four critical needs identified by analyzing the data of individual caregivers' experiences in stroke recovery. The four needs are information, involvement, support and self-care. **S1 File** presents the identified needs and describes individual requirements for individual needs.

### Need 1: Information

Most caregivers emphasized the importance of information to reduce uncertainty and be better prepared in stroke caregiving. Due to the sudden onset of stroke, most caregivers find it challenging to understand its occurrence, leaving them shocked and helpless due to this experience.

> *"It was very hard to come to terms with what just happened. . . there was no warning."* [Social Media]

> *"I don't know how to get through these dark days. . . I feel isolated and alone"* [Social Media]

Due to the uncertainty followed by the onset of stroke and the inability to understand the cause of the disease, many caregivers often searched for an explanation to cope with the disease.

> *"Frankly speaking, I'm quite confused now, because it is a sudden thing. . .ya, of course, this is my first priority (to know more about the condition). . .we have to learn a lot of things? I don't know"* [21]

Most caregivers relied on healthcare professionals such as physicians, nurses and therapists for information to help cope and prepare them to navigate through a 'completely foreign world' [22]. However, medical professionals primarily provided the stroke survivor with information, with little consideration to the caregiver. Moreover, some caregivers reported poor physician service attitudes during post-stroke discussions with the clinicians [23].

*"I don't think they told me anything, I was just left out in the cold. . . I didn't have a clue what was going on . . ."* [24]

*"His attitude was somewhat arrogant. He won't answer everything that we asked. I don't know if it is because of our manner or what. . . I feel he did not tell us much information. We need to try on our own to understand it."* [23]

*"I know as a family member or a patient we have the right to know about what the treatment is for and the medication for. You should give me an explanation. But he didn't. Every time when I asked him, his attitude was always like this way. . .Then he answered me, "Are you the doctor or am I? You don't trust me*?!"* [23]

Despite the limited information and communication with the healthcare professionals, a few caregivers mentioned that the information provided to them by the healthcare professional was established based on brochures and pamphlets [25, 26], which were not particularly useful [21]. One caregiver mentioned, *"Honestly, a layperson might not be able to understand those leaflets and they won't read them! So I think the leaflets are not worthwhile"* [23]. The lack of information provided to the caregiver during stroke recovery left the caregiver distressed.

*"They [medical professionals] weren't forthcoming with information of things to do. . .you feel like you don't know what to do. . .it felt pretty helpless"* [25]

*"With the lack of information, I really don't know what to expect."* [Survey]

Most caregivers discussed the need to have sufficient information on topics related to the disease, cause, effects, signs and symptoms, risk factors, prevention techniques, healthcare procedures, home care procedures, treatment and recovery options, locally available services, medications, and aid/tools as shown in **S1 File**. The caregivers preferred the information to be delivered clearly in 'layman terms'.

*"I was told by the hospital that they are removing him [stroke survivor] from the high dependency ward to the general ward where beds were available. I am not sure what this means. I need someone to explain to me in layman's terms what this means."* [Social Media]

*"My wife and I care for my grandmother, and we have no healthcare background, which has resulted in a very tough learning curve with the information provided to us."* [Social Media]

The caregivers also mentioned the importance of timing of information post-stroke, as the information needs are said to change over time [27]. This is because caregivers found it easier to recall information based on their current setting than provided at the time of stroke or discharge [25]. Further, caregivers also discussed the need to have personalized information delivered in several different ways based on their levels of literacy. Some examples discussed by the caregivers for the delivery of stroke information include (i) oral discussions, (ii) visual and written methods, and (iii) hands-on training by healthcare professionals. Oral discussions and hands-on training were particularly valuable methods for information delivery. They allowed

the caregivers to generate skills at the beginning to help support the person affected with stroke.

> *"My preference would be the first-hand experience. I mean to have the nurses, doctors, and medical staff actually sharing with me: 'Your father has stroke, so the best thing that you can do now is to. . .' Preferably, they can share some advice on things that I can carry out at home to look after my dad. . ."* [21]

> *"these [care practices] are not difficult. They just teach me one time then I can do it"* [23]

> *"They played a videotape for us and some papers. . .such as how to prevent a fall, how to care for a stroke person. . .but I still don't understand after reading those papers and watching the videotape. I needed a real situation to increase my understanding"* [23]

Besides oral and verbal discussions, caregivers preferred to receive visual and written information if someone explained it more comprehensively. Furthermore, visual and written information allowed the caregivers to access the information later should the need arise with no other form of support available [22].

> *"They do have some brochures and pamphlets but those are not really—I won't say not up-to-date—but those just explain the stroke condition very basically. . . What I reckon would be really helpful is to have someone brief us and go through them with us."* [21]

> *"It would have been nice to have somebody sit down with me and say this is what's happened, this is why it's happened, this is what you can expect. Okay, so it is there in the leaflets but you just kind of flick through the leaflets"* [24]

> *"But the fact that she, instead of just handing me information, she sat down and went through each point even though I was so tired and I'm going to remember it so much more. When I need to, I'll know in the sheets that she gave me where to go to look for the information."* [27]

## Need 2: Involvement

Caregivers discussed the need to be actively involved in the care process, which includes understanding the disease and practices to support the survivor in their recovery. The involvement in the care process added to their sense of preparation and confidence. Moreover, it would allow the caregiver to ask questions leading to a good learning experience instead of providing the clinicians with progress updates.

> *"I think it's good for the family to be involved and to know what's going on. I realize that when you are dealing with the public some of them don't know what you're talking about and some of them don't care. But I think that the majority of us want to know what's going on . . . and if we don't understand it immediately, then we need to have it explained to us in some terms that we do. We all need to be involved in our own health care, and we need to be involved in whatever needs to happen for people for whom we're taking at least some responsibility for"* [28]

Some caregivers expressed that they saw the caregiving role for their parents as an opportunity to get involved in the care as they received when they were a child. While some caregivers did it for religious beliefs or lack of confidence in the survivor to take care of themselves. Hence, leading to a significant emotional impact on the caregiver's decision to provide care.

*"I just thought that she had looked after me since I was a baby. When she can't help herself, I have to help her."* [29]

*"I feel that it's my duty. No one is free to take care of her, so I have to do it. If I get a job or become more stable than this, I might hire someone to take care of her. I am 28 years old now. I need to go to work. I feel that it's my duty to look after her so I do it. In my opinion, no one can do it better than me."* [Social Media]

*"I can't stop myself sometimes in letting her know when she makes the error. It is just to help her understand that she still does need help for things."* [Social Media]

In addition to the emotional factors, caregivers felt that by being more engaged in care they would be able to get reassurances of their fears and worries through encouragement derived from feedback from medical professions during stroke recovery [27]. Moreover, it would allow the caregiver to effectively communicate with the healthcare professional to make informed decisions in the best interest of the caregiver and survivor.

*"By being involved in care I feel reassured. I can engage with providers discuss my fears and worries and share with them the needs of my loved one. This allows me to carry on in my daily activities and feel a part of the recovery."* [Social Media]

Despite caregivers expressing their needs to be involved in care, most caregivers feel left out. Caregivers expressed their interactions with medical professionals as uninvolved or disconnected [10]. When approached by caregivers, medical professionals either did not provide enough information or expressed that they were "too busy", which left them uncertain [10].

*". . .but we do not get much information [about] his condition. There is no update. You see, I have to go and ask them. They say, 'The doctor will come but don't know what time.' Do you understand?"* [21]

*"The communication between me and the hospital leaves a lot to be desired. Every day I wake up and I'm still in the twilight zone."* [Social Media]

The uncertainty led to numerous concerns regarding the rehabilitation received by the survivor and the caregiver's own preparation to assume the full-time role as a caregiver. One caregiver mentioned *"Well one of the things that's uncertain is not knowing when she's coming home and also not knowing what level of care she'll need"* [10]. This led to the caregivers developing negative emotions.

*". . .we're still very shocked. . . we're trying but we're not coping well. . .our family is going haywire. . ."* [Social Media]

*"I am scrambling to get a caregiver. . .I am so stressed. . .In fact, we are very worried. . ."* [Social Media]

*"I'm probably still coping, but I still get stressed up sometimes. . .because he's never been so sick. . .all of a sudden. . ."* [Social Media]

*". . .at least he can eat. . .if he needs tube feeding, then we're in trouble. . .might need to send him to a nursing home."* [21]

Furthermore, the lack of communication with healthcare professionals led the caregiver to feel less supported therefore contributing to frustration.

*". . .the hospital never answered my questions about care package, they didn't tell the home that my mum requires one on one care—they also didn't tell us that she can get up and walk on her own. Her speech is very muddled, but some make sense. I asked the hospital three times over a two week period to provide me with details of the care package and they never got back to me. The home has called emergency doctor out, and she's said that my mum shouldn't have been brought to this home, as clear she needs further physio, speech and ot"* [Social Media]

*"I have found the lack of treatment pretty appalling. . . I do feel quite angry. . . she needs more treatment. . . she could make far better progress if she were getting better treatment."* [30]

The lack of preparation and communication was not the only reason for the caregivers' involvement in care. One caregiver mentioned that the survivor would constantly 'push' them away during stroke recovery. While another caregiver mentioned that her husband was 'refusing treatment'. The lack of motivation amongst the survivor causes friction with the caregiver leading to depression and withdrawal from care.

*". . . he [survivor] has been going to speech therapy off and on. . . He is stating that he does not want to go anymore. Typically, when he doesn't want to do something, he is pretty ornery and stubborn about it. Should I push the issue?"* [Social Media]

*"Sometimes getting along and sometimes arguing, arguing just because he doesn't want to do these exercises . . .and he doesn't want to do it he just sits there and I say come on let's go let's get up let's do something. . .and I feel so frustrated sometimes and I get angry and that's what we argue about. . .."* [31]

Regardless of these issues, caregivers have provided care to the survivor in planning, caring, managing, motivating, and goal-setting activities. These activities are critical in ensuring the survivor feels supported throughout stroke recovery. Caregivers have employed a range of strategies in their efforts to care for the survivor. For example, some caregivers considered taking one day at a time and did not plan for the future. In contrast, other caregivers created 'meaningful goals' for the survivor to create ongoing reassurances of their recovery. Furthermore, the caregivers also set up medical and follow-up appointments and participated in physical therapy to add to their sense of preparation and confidence.

*"I train her cognition and memory. I write some words or numbers and show it to her. Such as I write our four kids' names on four cards then show her first. Later I pick one and ask her what's the name on it and which child's name. . ..things like this. Those strategies pop out of my mind. I use those skills to train her cognition"* [23]

*"I often talked with her, very often. . .. I played with her the childhood games she taught us. And I sang her favorite songs to her. . .. One day I asked her: Mom, do you want some Zong Zi (a pyramid-shaped mass of glutinous rice wrapped in bamboo leaves)? She answered, "No". . .And on July 4th, she could say my aunt's name. Now she can sing her favorite songs with us. . .At first, she only can sing in her mouth, but now she has the voice. She cannot recognize us, but she can sing those songs!"* [23]

*"I faced him and put his arms on my shoulder; then I walked backward to guide him to walk. His left foot could not move, so I used my right foot to hook his left foot and pulled his leg forward. I did this during his way to the bathroom and back to his bed."* [23]

Caregivers in stroke also discussed methods to be considered to be more involved in care. The primary consideration is for healthcare professionals to consider the caregiver as the 'central role' in the survivors' rehabilitation [32]. This can be achieved by creating a close collaboration with clear communication between the caregivers, patients and healthcare professionals. The preferred mode of communication amongst caregivers was face-to-face communication to build a trusting relationship, which may be critical during the counseling process [33]. For example, one caregiver mentioned, *"I need someone who gives me the time I need and where I can also show emotions"* [33].

Other than efficiently communicating with the healthcare professional, the caregiver preferred being a part of the decision-making process for the next step of rehabilitation and receiving training from a healthcare professional to provide improved care. The inclusion in the decision-making process allowed them to share their opinions, such as the type of care, availability (or time) to attend care and survivors' preferences for care. By ensuring tailored or individualized care, caregivers felt more satisfied and confident in their ability to support the survivor. Moreover, it was considered to be 'very important in maintaining stroke survivor health'.

As for care training, most caregivers stated that they would 'require training to help their family member specifically those skills needed for transfer and at home after discharge. One caregiver explained, *"I felt so hard because I never bumped into this before. I never faced a seriously sick person. This is my first time. I had no idea what I have to do for her. . ..but if I want to take care of her, I must learn"* [23]. The caregivers believed that hospitalization would be the best period for them to learn from experts (i.e., healthcare professionals) and gain hands-on experience. Moreover, it could allow them to gain some feedback during care.

*"if my father is referred to the rehab centre, I could probably be there to watch them and learn a lot. . .on-the-job training."* [Survey]

*"the healthcare professionals can help but they are so busy."* [Survey]

*"The community hospital will train the maid at the bedside and ask her to do [the care tasks]. She [survivor] will need to stay there for one to two months. That should be enough time to train the maid."* [21]

*". . .we used the chance to learn. We were in learning process, so we had to do hands-on it. If we learned it, we might feel less pressure. We could not start to learn it after going home. So in the hospital, if we are unable to do it, we can ask for help from them [healthcare professionals]"* [23]

## Need 3: Self-care

The physical and psychological impact of caregiving was common amongst most post-stroke caregivers. Caregivers reported having feelings of anxiety, loneliness, fatigue, frustration, burnout, fear, social isolation and helplessness during their role as a caregiver due to the alterations of their lifestyles, life status and routines in a negative way.

*"Ok I admit I am defeated. . . I can't keep the happy me anymore or keep up with everything. Maybe just tired or trying to deal with too much, but I do not have a way to fix it or to even try to make a way to make it work. I have bitten off more than I can deal with. . . I need to be*

*here & go to work & also get chores done & also be able to try & get me time. It's just not possible.*" [Social Media]

"*[I am] exhausted, drained & overwhelmed feeling. It's a lot to take in!!*" [Survey]

Most caregivers felt that caregiving's physical and psychological impact was the lack of independence to engage in other activities that previously ensured the fulfillment of life. Before the stroke, caregivers participated in activities such as work, social gatherings, volunteer work, religious activities, vacation and trips, which were halted due to the dependence of the survivor on the caregiver.

"*Since he's [survivor] come home I've not really gone out very much. Normally I would just go out and do whatever, but I haven't been able to do that since he's come home from the hospital.*" [24]

"*I've quit doing gym. I have virtually given myself up. I only live for my mother and my husband here at home, and don't do anything else.*" [34]

Caregivers often put their dreams on hold as they were responsible for providing constant care for the survivor. For example, one caregiver decided not to commit to future employment opportunities, while another caregiver couldn't go on a vacation with their loved ones. The reason for putting their 'lives' or 'dreams' on hold was the survivors needs and uncertainty during recovery. However, giving up their dreams was not always done readily [35].

"*We put a lot of our dreams on hold, because we don't see how we can go on vacation. We think about who can watch our loved one, or how can I take him/her [survivor] with us.*" [Social Media]

"*I've been angry. I resented having to quit the full time job where I was making good money and I loved the job. . .I didn't want to have to do that*" [35]

Work and social participation were two critical themes discussed by caregivers to ensure fulfillment of life during their care journey. Currently, most caregivers have either given up their work or have reduced their work hours to accommodate survivor care. Caregivers who have quit their jobs feel 'homebound' or 'isolated' [36, 37], while those who have managed to continue their work report having difficulties in maintaining work and care responsibilities.

"*this [work] has taken a big toll on me–I am juggling between my job, my husband and children, my husband's business and caring for my mother–sometimes I can't see any light at the end of the tunnel*" [37]

In terms of social participation, stroke has altered the lives of the caregivers in such a way that they can no longer participate in the activities that give them 'joy' or 'happiness'. One caregiver discussed "*not having enough time for himself*"; while another caregiver mentioned that they felt "*restrained from going out because of the caring activities and housework*" [38]. The inability to participate in social activities strains the relationship between the caregiver and the survivors due to the mental and emotional aspects associated with being isolated at home and the loss of independence.

"*I used to have lots of outing activities eight or nine years ago. Now I feel so alone*" [38]

*"The kids used to love camping, but now they have to go with friends, my wife can't do that. They also love going to the movies but my wife is sensitive to light so we don't go anymore. One of us has to stay with my wife at all times, we take turns. I miss the spontaneity; the stroke restricts our lives. . ."* [26]

*"My dilemma is finding me time without making them feel left out, which is what's happening. It's straining our relationship because of the mental and emotional toll. . . I'm trying to be compassionate, forgiving and understanding; to see things from their perspective but I'm almost burnt out."* [Social Media]

Caregivers also discussed practices currently employed for self-care. The practices involved: (i) distracting their partners by tempting them to engage in other activities, (ii) participating in relaxing activities such as cooking, going to church or other household activities, and (iii) taking time off during the survivors' rehabilitation appointments. Caregivers believed that these practices allowed them to 'take a break' or 'get time for themselves'.

*"And I do [activities of my own], because my activities also help me get distracted, because it happens that I stress myself with my mother's illness. It is worse, when I get sick, though. So, what I do is do my normal activities as I did before"* [39]

Despite developing practices to ensure self-care, caregivers often report wanting more time to engage in leisure and self-care activities, which has contributed to poor health. To further improve self-care practices, caregivers mentioned potential considerations to improve physical and psychological well-being. Caregivers discussed the need to take breaks from physical care, and either 'going back to work', 'travel' and/or 'participate in recreational activities' as illustrated in **S1 File**. Some described this situation as 'out of sight, and out of mind'. However, it was only possible for the caregiver to take a break when they were able to share responsibilities with other family members and/or friends during recovery.

## Need 4: Support

Caregivers discussed the importance of receiving support–in terms of resources, services and finances to sustain their ability to provide quality care (**S1 File**). As caregivers mainly were family members or friends of people living with stroke without prior experience in providing care, they expressed the need for help or support, especially from the healthcare system.

*"the system could have provided more psychological and emotional support; also a long-term follow-up of how we are doing would be important"* [37]

The lack of care support can affect the caregiver mentally, leading to uncertainty, fear, loneliness, anxiety, anger and frustration leading to financial and relationship impact.

*"How am I going to cope?. . . I don't know I really don't know. . . when the worry is on my shoulders, that's all I'm thinking about. . . I don't think I could take the worry. I don't— maybe I'll have to."* [30]

*". . . I am trying to stay positive for him [survivor] but I don't have any answers and I am feeling so alone and anxious. I just want to know it will get better."* [Social Media]

*"I feel so overwhelmed at times, feel like I'm doing it all wrong,when it comes to important stuff like our finances or dealing with Dr. or the government I feel so inadequate. just scared*

*I'm gonna screw up something important that can't get fixed. Only been doing this for 8 months. Not getting much help."* [Social Media]

Caregivers discussed that uncertainty was due to the lack of understanding of the disease and future life. For example, one caregiver mentioned, *"And what the future will bring nobody knows. Sometimes I think it can come back, because they told me, it could happen, another stroke. Because most people come back with another stroke"* [40]. Further, the uncertainty towards the future could result in fear in terms of recurrence of stroke, falls, finance and relationships. As a result, most caregivers choose to be at home to observe and care for the survivor, contributing to isolation and loneliness.

*". . . now I have to hurry even to go to the supermarket because I'm afraid. . . to leave him. . . He may have a CVA again at any time, as the doctor said. Like the first time it happened. He fell to the floor, urinated on himself, he fell in the middle of the night. He fell off the bed, and I didn't even notice. So, I'm afraid. So, I hurry to supermarket, but my mind stays here . . ."* [34]

*"I am struggling with feeling very alone. My husband had a stroke in May and it's getting to the point where I don't feel like a wife or partner anymore. I'm a caregiver 24/7."* [Survey]

During caregiving, caregivers have also discussed feeling of anger and frustration, which is contributed by issues related to care, financial, relationship, emotional and social hardships.

*"Sometimes I think I may be going crazy! We are going on almost 3 years post-stroke and it's so hard to deal with still. I lose my temper so easily now, full of anxiety and anger which I end up taking out on my husband who has had a stroke. I feel so bad after it happens and it's not any particular thing that sets it off. I'm supposed to be the strong one for us—hes' the survivor —I'm the one angry and emotional all the time now?? It's almost as if I'm experiencing his symptoms if that makes sense. I should probably speak with someone but I have such crappy insurance and can't afford another Dr. bill. . . ugh!"* [Social Media]

*"Ever get to the point where you have so much on your plate that you cry and laugh at the same time and then have nothing but oreos for dinner and want to throw your hands up and scream at the universe to back the f\*\*k off?!Aaaand I go back to my wine."* [Social Media]

The financial impact is a common stress factor amongst caregivers. Most caregivers have stopped working altogether to support their loved ones, which made them entirely dependent on financial support from other sources such as family members, insurance, pensions, etc. Some caregivers described that these sources of support were not sufficient to cover the survivor's expenses and as a result resumed working. However, this contributed to increasing workload, caregiver burnout, and guilt due to the inability to provide continuous care.

*"This [the financial situation] has most affected me, because now I have to wait for my son to give me money, while before I always solved all of the problems related to the household on my own."* [39]

*"My God, what will my life be like now? I won't work."* And then I started thinking: *"What will I do if I don't work?"* [Social Media]

Caregivers also discussed the impact of caregiving on their relationships with the survivors and other family members. For example, one caregiver mentioned, *"I wasn't able to be there as much as I really, really wanted to be, because obviously I was seeing to [partner's] needs. . .we*

*would have had a much more hands-on relationship with the four grandchildren...*" [32]. The caregivers' inability to maintain relationships was due to their failure to balance the different roles. One caregiver discussed that their partners were upset with them for being the second priority in their relationships and would often "complain" about the time and effort taken by the caregiver in caregiving. While another caregiver described that the lack of support received from their partner in the caregiving role resulted in them working for "two people", which resulted in friction or conflict and thereby increasing their burden.

> *"With my partner who is far away, that's really hard because he feels like he's second most important. . .. because a lot of my time is spent at home with my parents and helping my dad. So that's kind of suffering"* [35]

> *"At the beginning, when my father got sick,. . . I argued with my husband and he left. We were separated for almost a year. Then, he came back. Then, he decided to separate again because I was too nervous. He even said that he could understand me, but couldn't take that situation anymore... It's difficult! You see. I had my house, my life there. My husband and I. Then, we left there and moved here, thinking that things would be a certain way, and now that we are here, he has to be everyone's father . . . he's playing a role that shouldn't be his. My husband is a very good man, but who can take all this?"* [34]

> *"All that being said. . .we have been raising our 3 grandchildren as well. They have been a handful for me for 6 years, all being neglected in the past. So they are special needs as well, each with different needs. They are state kids, wards of the court, and we fought for them to come to us instead of being separated and adopted out, or placed in-group homes/foster homes. Majority of the work fell on me, counseling, doctors, court meetings. . .etc."* [Social Media]

> *"There was maybe just a little bit of tension initially with my children because they're small and don't understand the seriousness of the situation and I guess they couldn't understand why I always had to go to the hospital"* [35]

The impact of caregiving, i.e., the mental, financial, relationships and care described above, can obstruct recovery. For example, caregivers with financial difficulties explained the inability to afford rehabilitation or community services to improve survivor recovery, while caregivers with relationship and mental hardships discussed the issues in engaging in care of the survivor.

> *"We hired a person who performs exercises. Rs. 1000 per day. Later, we had to face some financial difficulties. so we couldn't continue that."* [36]

> *"I guess from my perspective probably the friendship that is most lacking is the friendship with him. That's–(now I'm teary sorry)–it doesn't exist. No it's, I have become nothing more than a carer, you know–the secretary that sits at the computer typing, doing his emails, reading everything to him, telling him about everything that happens. But that's all there is to it"* [31]

> *"My mom had a stroke almost 1 month ago and is now in rehab. We lived in the same town. My brother lives in 2 provinces over and came when she had a stroke. We live in Canada. We are her power of attorney. Can't agree on anything. And now my mom seems to be agreeing with him and I'm being pushed out! I don't understand. . ."* [Social Media]

As a result, caregivers often look for support to ensure better care for their loved ones, and limit their burden (or burnout). Currently, some caregivers have discussed receiving financial

and care support from relatives. Relatives assisted caregivers, including financial aid, visitation, food, advice, materials and equipment, engaging in treatment, and occasionally supporting the caregivers at tasks. Caregivers found that receiving support from relatives could prevent loneliness and isolation and allow them to engage in social activities, work, self-training classes, volunteer groups, and travel. One caregiver stated, *". . .my daughters and my sons-in-law have helped me. They are very good to me. Our family is very attached. Whenever I need it, I call, and we can always manage to arrange schedules, even though they have a job"* [34]; while another caregiver discussed the opportunities bought about by relative support *"Having more caregivers other than family members [would help]. . .because it would give us, the family members, an opportunity to get away from the parent. . .give us some time that we can go out and have time on our own. . . Give me a break!"* [35]. However, not all caregivers received support from relatives to care for the survivor. These caregivers found it challenging to adjust to post-stroke conditions. One caregiver stating that *". . .our children have been less supportive than. . .expected. . .because the ball game has changed for them too, there is no-one around to support them, so they have been of less practical assistance"* [9]. These caregivers often decided to live alone and relied on friends, neighbors, and healthcare providers to ease their burden. In addition to relatives and friends, caregivers often drew on the experiences of other caregivers to cope and understand the situation.

> *"I know from another group I'm in that other members are wonderful about giving advice and I hope I can learn from them, and maybe offer some advice of my own."* [Social Media]

> *"being in a group [support] of individuals with similar experiences provides me with comfort and hope, and abundance of information for which I am grateful to God."* [Social Media]

> *"ya, I think it will definitely help me, because um. . . some of them may have the experience to be caregivers for stroke patients for many years. For me, I'm just starting out, I'm very new. Definitely there will be lots of mistakes, and my learning curve will be like ups and downs. So definitely if the more experienced caregivers will share with me advice or tips on how to look after a stroke patient. Definitely like I will find it very useful. Like, for me, I need a mentor to become my buddy you know. I'm really very blur, I do not know what to do."* [21]

Caregivers often found it challenging to find caregiving peers and support groups, and often discussed meeting other caregivers on internet-based support groups and outpatient waiting clinics. One caregiver stated, *"My Dad just had a TIA Wednesday. He had one three weeks prior also, but never went to the hospital. This time I made him go. All tests came back normal, which to me is concerning cause then they don't know why these are happening. It makes me so nervous about another TIA or even worse. So, I'm happy to be on this online support page for him"* [Social Media].

Other than learning from the experiences of others, caregivers often found it crucial to have coping strategies. Coping strategies allowed caregivers to adjust to their new roles and stay 'strong' and 'positive' [40]. Self-encouragement, strong determination and self-strengthening of the mind were considered critical strategies to cope with the caregiving role [36]. Another coping mechanism discussed by caregivers was including relaxing activities such as praying, cooking or participating in leisure activities that allowed the caregiver to feel independent.

> *"[. . .] I sit down and start to pray and say "please, the life of my husband is in your hands. I don't have faith in anything else other than in You. You will give me enough courage," and I pray for half an hour, ten minutes, five minutes, the time that I have, then [. . .] afterward I feel better to go on facing the problems I have. This is my way, it is not, as I told you, it is not*

*that I pray and then the things get resolved, to resolve them, I have to resolve them myself."* [Social Media]

Further, the caregiver frequently relied on acceptance and patience, the release of temper, rest or relaxation, self-treatment, time management, seeking and receiving help, letting go, using materials and equipment and creativity to manage complex issues with care.

*"I think we have all learned to adjust. . .that's a big word because our lives changed drastically when it happened. We have to learn to adjust and accept."* [9]

*"Just the fact that [survivor] wasn't able to do what he had been able to do previously. . .acceptance was a big thing in that regard"* [9]

*"You couldn't return to where you were. . .people think that getting better is getting back to as you were. She got better but in a different way. We evolved our life in a different way"* [32]

*"Things are good, getting better all the time, it's not stopped. We're not in the situation where we're going to be like this for the rest of our lives. We've still got our lives to live and we will"* [32]

While caregivers found creative methods to deal with the burden of stroke caregiving, they often looked for support from the healthcare system. The type of support valued by these caregivers for themselves included information, emotional, social and financial support.

*"We did not get as much help as we needed–I could have used more respite, especially at the beginning, my questions were poorly addressed by the [local community health clinic]"* [37]

*"this is difficult financially because the supplies are very expensive, and the insurance doesn't cover very much"* [Survey]

Caregivers also looked for current healthcare services that were tailored to their needs. The needs included flexibility in visiting healthcare staff or rehabilitation programs, as this allowed for them to a break from their caregiving duty for a few hours each day.

In addition to receiving support from healthcare systems, caregivers also highlighted the importance of informal support. For example, one caregiver demonstrated the ability to take refuge in family members *"It does help the fact that, that I know that what I feel, he feels exactly the same way. . .I phone up and moan at him about things and feel better for getting it off my chest and he phones me about things, and I know it makes him feel better, but nevertheless for both of us it is still difficult to cope with things."* [30]; while another discussed the importance of supportive and understanding relationships during the care journey.

Caregivers also stressed the importance of financial, social, emotional, and physical support. Financial support was critical for caregivers to provide efficient care and would allow caregivers to remain less uncertain about the future as described by one caregiver *"Hardest part is cognition and him [survivor] not wanting to let go of me—he fears/anxious about being alone, but I HAVE to return home to hubby and my job! Can't keep up the overdrawn bank account"* [Social Media]. On the other hand, social and emotional support is essential to support the vulnerability of the caregiver as it enables the caregiver to 'master their fear' of caregiving and ensures they 'feel cared for by others'.

*"My daughter helps me–she calls every day to see how I am doing because she knows this is very hard for me and she has us over to dinner once a week"* [37]

*"I mean, we [siblings] have always been very united,[. . .] we are one; this situation connected us even more,[. . .] in life money is not everything, not everything in life is about having something. Life is living in harmony, and this has united us more. . ."* [39]

*"I have some help from friends to keep up my yard. . . most people don't understand what I am going through or how much energy it takes to look after someone who is not well"* [37]

Some caregivers expressed difficulties in managing the physical issues of caregiving, especially those that require physical strength, i.e., moving or carrying the survivor. Hence, caregivers often looked for materials and equipment that can help reduce the stress of physical activities, especially during emergencies. However, most caregivers expressed a lack of proper understanding of the tools essential for recovery and local services available to support the survivor. Hence, requiring additional support from healthcare professionals to ensure limited physical stress on the caregiver.

Caregivers also expected healthcare professionals to assist them in gaining skills to 'pacify' and motivate the survivor. One caregiver mentioned, *". . .is there any possibility that any plan is drawn up, to really let the patient—instead of us (the family)—know what he may expect now-. . .if they are still conscious of what's going on, I think it will be good to tell them (stroke patients) what they are or what they will be going through"* [21]. The ability to gain skills to support the survivor would allow the caregiver to remain 'positive' and 'hopeful' about the future.

## Discussion

Caregivers of people living with stroke at an increased risk of negative health [41] as a consequence of the lack of preparation for managing and supporting the unexpected and complex nature of stroke recovery [42]. While many caregivers have attempted to manage and support the needs of people living with stroke through the formulation of strategies. Our findings suggest that this alone does not reduce the overall caregiving burden and quality of life. Hence, requiring a clear understanding of the needs and experiences of caregivers of people living with stroke to ensure healthcare professionals and researchers are aware about the support requirements of these individuals. It is when the needs of the caregiver of people living with stroke is addressed that they can feel a sense of satisfaction with their role [43], and in turn improve not only their own health and well-being but also contributed to the reduced use of healthcare resources by the person living with stroke [44].

### Comparison with literature

This grounded theory explored the needs of a caregiver of people living with stroke described based on four key themes: information, involvement, self-care and support. The contributing factor to these needs was the 'lack of preparedness' due to the sudden onset of the disease. A consequence that has been described several times in the literature [45–49]. While information was considered a 'critical' factor towards improving preparedness; several caregivers valued being involved in the recovery process of stroke and having support from both healthcare professionals and other stakeholders. Despite several studies describing these needs, the burden of providing care to the person living with stroke is still prominent.

The main difficulties in addressing these needs are the 'lack of communication' and 'lack of support'. It is known from our findings and the literature that the caregivers of people living with stroke are dependent on the healthcare professionals to provide information and support services. The information preferred by the caregiver is expected to be personalized to individual situations and health literacy; with a majority of caregivers preferring visual demonstrations. The importance of visual demonstrations for the delivery of stroke information is not

new, and has been discussed previously in a study by Lobo et al. [19] as a preferred method to ensure maximum interaction. However, our findings show that the information provided is in the form of generic pamphlets and brochures with limited explanation from the healthcare professional that led to confusion. This lack of communication from the healthcare professional regarding the disease and methods to access support services has shown to contribute to increased frustration, burden and a sense of loneliness amongst the caregivers as indicated by both our findings and the literature [50–53].

Our findings highlight that the 'lack of communication' is due to the healthcare professional being 'too busy' to engage with the caregiver. In addition, the literature highlights that the caregiver of people living with stroke often felt the healthcare professional 'lacked interest' in communicating with them [54]. Overall, contributing to the limited trust the caregiver has for the advice provided by the healthcare professional [55]. It is, therefore, important for healthcare professionals to actively engage with the caregiver in the recovery process through collaborative interactions [10].

Apart from considering the needs of the caregiver in providing care and support to the person living with stroke; our findings indicate that caregivers also described the need to support their own health and well-being and quality of life. With the onset of stroke, many caregivers have to restructure their lives to accommodate the care needs of the person living with stroke, including adjusting their household environments, work schedules, living arrangements and everyday routines [56]. The process of restructuring of the lives of the caregivers has shown to contribute to a variety of experiences and feelings [57]. In a study by Visser-Meily et al. [58] it was described that more than forty-five percent of the spouses of people living with stroke had experienced high levels of depression, burden and dissatisfaction with life after 1 year of stroke.

Our findings highlight that most caregivers often want to engage in self-care by participating in work and leisure activities, with an intention to take a 'break' from their caregiving role. This often requires support from other stakeholders such as family members or friends to assume the role of caregiver while they can engage in these activities. Similar findings was seen in a study by Steiner et al. [59] that describes the importance of the caregiver to establish am adequate self-care system that considers physical help and emotional support from family and friends to protect the caregiver against poor health outcomes.

While in this study we have described the four critical needs of caregivers of people living with stroke. It is important to note that the needs of the caregiver change over the process of recovery [33]. For example, during the first few weeks at home the needs are more related to the practical aspects of providing care, while over longer periods the focus is towards community reintegration [60]. Hence, demonstrating for a need for change in delivery of healthcare in stroke recovery [61], with focus towards providing a more personalized, flexible and available support as much as required by the caregiver of the person living with stroke [62].

## Strengths and limitations

The study was conducted in three phases, which include: (i) literature analysis, (ii) social media analysis, and (iii) caregiver survey to identify and evaluate critical needs in stroke caregiving. Each of these phases is not without its limitations. In Phase 1, the search strategy focused on articles published in English language, which may have resulted in the exclusion of articles relevant to this study, and that may have presented a different set of needs for the caregiver of the people living with stroke. In Phase 2, the social media analysis poses several ethical concerns. For example, while the research focused on publicly available posts available on social media it does not specifically request consent from the individual to access or analyze

the data. Another ethical concern is due to the vast number of users accessing social media and posting their needs online, it was difficult for the authors of the study to contact each user to determine their authenticity. Hence, considering to be a critical ethical issue. In Phase 3, the focus was on using online surveys rather than face-to-face interviews towards creating the core categories and storyline for the grounded theory. This was as a consequence of the restrictions imposed due to COVID-19 and the limited acceptance of caregivers of people living with stroke to use video-conferencing technologies. Face-to-face interactions with the caregiver may have allowed for a more detailed understanding of the caregiver's needs, which may have affected the richness and quality of data collected. Despite these limitations, all findings presented in this study have undergone rigorous analysis and documentation to ensure the caregivers' needs are well represented with specific consideration to the anonymity of the caregiver.

## Implications for future research

The findings in this study have several implications for clinical practice, healthcare planning, and healthcare policy. In particular, this study sheds light on the needs of stroke caregivers across stroke recovery. Thus, considerations are made to address these needs to bring about preparedness, reduce uncertainty and promote behavior change. One plausible approach is through the active engagement of the caregiver by the healthcare professional.

A study by Lobo et al. [61] describes that to facilitate active engagement, caregivers often need to be informed about the disease, rehabilitation and decision making processes. The influence of actively engaging the caregiver in the recovery process is supported both in our findings and in the literature. For example, a study conducted by Krieger et al. [33] showed that caregivers that were actively involved in the decision-making processes enabled caregivers to adjust to their new role, while also having a positive effect on the recovery of the person living with stroke. Additionally, caregivers who were actively involved in the rehabilitation processes [63] felt more prepared and had increased levels of satisfaction, while education has been found to influence the quality of life of the caregiver [64]. Hence, making it essential to consider in future studies looking to address the needs of stroke caregiving.

## Conclusions

The study drew on grounded theory to explore the needs of caregivers of the people living with stroke based on their experiences in care. The core categories identified, i.e., information, involvement, self-care, and support, to empower the caregiver while reducing uncertainty, mental impact, financial stress, and burnout. In particular, it was clear that there are several barriers in the current healthcare system leading to unpreparedness and unwanted effects on the caregiver. Hence, requiring the inclusion of a more caregiver-focused approach in clinical practice and health care planning.

## Supporting information

**S1 File. User needs and associated comments.**
(XLSX)

## Author Contributions

**Conceptualization:** Elton H. Lobo, Anne Frølich, Mohamed Abdelrazek, Lene J. Rasmussen, John Grundy, Patricia M. Livingston, Sheikh Mohammed Shariful Islam, Finn Kensing.

**Data curation:** Elton H. Lobo, Anne Frølich, John Grundy, Sheikh Mohammed Shariful Islam.

**Formal analysis:** Elton H. Lobo, Anne Frølich, Mohamed Abdelrazek, John Grundy, Sheikh Mohammed Shariful Islam, Finn Kensing.

**Investigation:** Elton H. Lobo, Mohamed Abdelrazek, Sheikh Mohammed Shariful Islam.

**Methodology:** Elton H. Lobo, Finn Kensing.

**Supervision:** Anne Frølich, Mohamed Abdelrazek, Lene J. Rasmussen, John Grundy, Patricia M. Livingston, Sheikh Mohammed Shariful Islam, Finn Kensing.

**Validation:** Elton H. Lobo, Anne Frølich, Mohamed Abdelrazek, Lene J. Rasmussen, John Grundy, Patricia M. Livingston, Sheikh Mohammed Shariful Islam, Finn Kensing.

**Visualization:** Elton H. Lobo, Anne Frølich, Mohamed Abdelrazek, John Grundy, Patricia M. Livingston, Sheikh Mohammed Shariful Islam, Finn Kensing.

**Writing – original draft:** Elton H. Lobo.

**Writing – review & editing:** Elton H. Lobo, Anne Frølich, Mohamed Abdelrazek, Lene J. Rasmussen, John Grundy, Patricia M. Livingston, Sheikh Mohammed Shariful Islam, Finn Kensing.

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
