## [Decision Letter · Decision Letter 0]

6 Jun 2022

PONE-D-21-28428

Information, Involvement, Self-care and Support - The Needs of Caregivers of People with Stroke: A Grounded Theory Approach

PLOS ONE

Dear Dr. Lobo,

Thank you for submitting your manuscript to PLOS ONE. After careful consideration, we feel that it has merit but does not fully meet PLOS ONE’s publication criteria as it currently stands. Therefore, we invite you to submit a revised version of the manuscript that addresses the points raised during the review process.

Should you decide to revise and resubmit your manuscript, please ensure that you address each point identified by the reviewers.  Be sure to explain what value your findings add to the knowledge base in this area.

We look forward to receiving your revised manuscript.

Kind regards,

Jeffrey Jutai

Academic Editor

PLOS ONE

Journal Requirements:

2. Please provide additional details regarding participant consent. In the Methods section, please ensure that you have specified (1) whether consent was informed and (2) what type you obtained (for instance, written or verbal). If your study included minors, state whether you obtained consent from parents or guardians. If the need for consent was waived by the ethics committee, please include this information.

Additional Editor Comments (if provided):

Thank you for submitting a manuscript addressing a timely and important topic. Should you decide to revise and resubmit your paper, please address each point identified by the reviewers.

Reviewers' comments:

Reviewer's Responses to Questions

**Comments to the Author**

1. Is the manuscript technically sound, and do the data support the conclusions?

Reviewer #1: Partly

Reviewer #2: Yes

2. Has the statistical analysis been performed appropriately and rigorously? 

Reviewer #1: N/A

Reviewer #2: N/A

3. Have the authors made all data underlying the findings in their manuscript fully available?

Reviewer #1: No

Reviewer #2: Yes

4. Is the manuscript presented in an intelligible fashion and written in standard English?

Reviewer #1: Yes

Reviewer #2: Yes

5. Review Comments to the Author

Reviewer #1: This is an interesting paper which aims to use multiple methods to synthesize and summarize the impact of providing care on caregiver health and wellbeing post-stroke. The paper is well written. Addressing the following comments may strengthen the message of the manuscript.

In the introduction, on page 9 of the proof, the authors highlight previous systematic and other reviews. Can the authors elaborate on what has been presented in existing reviews and what specifically is missing that this new project will address? Please follow this up in the discussion to highlight what we have learned from the current paper beyond what has been presented previously.

In the methods, the authors describe use of a grounded theory design with three sources of data (3 phases of data collection). The study design was actually more complex than just a grounded theory study. What is the rationale for using 3 data sources to address this research question? How do the 3 phases relate to grounded theory methodology? Can you provide more detail about the data sources in the context of grounded theory?

The first method used is a literature review. Can you clarify the type of review methodology that was used and cite this?

The methods could be strengthened by providing more information about the specific research phases. How were the data collected for each phase? What data was collected in each phase of this research? How was the data combined? At what stage, during data analysis or after? In the presentation of the results, the data source is not clear. Is there a way to indicate the source of the data for each of the quotations including the phase of the research where it was obtained?

Is there a citation for the “Python-based web-scrapping tool”? What is this tool and how does it work?

With respect to the phase 3 caregiver survey, can you provide more detail about the theory integration process and provide a citation for this approach? How specifically was this approach used? How was informed consent obtained? How were the questions asked? What types of questions were they? What were the psychometric properties of the survey? Please provide more details about the survey.

Table 3, participant characteristics, should include numbers and percentage in brackets. Clarify what is meant by relationship. Is this the person being cared for is a grandparent for example?

Please clarify how the data analysis integrated the 3 different sources of data. Please provide a citation for the data analysis approach – constant comparative analysis.

In the results, please add to the methods how the stroke disease trajectory was determined - from the data or existing source? if an existing source please cite or explain what is meant by the stroke trajectory.

Suggest starting each theme with a summary description of all of the dimensions included in the theme.

Please provide the sources of all quotations. This should include data type/phase, an identifying number

At the top of page 18, the authors suggest information needs to be provided at the right times, please explain. It isn’t clear how the selected quotation relate to this concept of timing. Clarify or select different quotations.

Towards the bottom of page 18 is this sentence “Further, caregivers also discussed the need to have personalized information based on individual needs based on personal experiences delivered in several different ways based on their literacy and health literacy levels” – it is complex, suggest revising to clarify meaning.

The authors suggest caregivers need “demonstrations” but the last quotation in this section suggests they need more than this, actual hands-on testing/development of their skills.

On page 20, Please further explain what is meant by involvement in the care process? Suggest this paragraph starts with an explanation / summary of the different aspects of involvement included in this theme.

On proof page 21, “Some caregivers expressed that they saw the caregiving role for their parents as an opportunity to give back as retribution for love and care when they were a child”, clarify relation to involvement theme. Is this related to the decision to be involved? Also, the word “retribution” – is this the correct word? It has a negative connotation.

On the top of proof page 22, “Moreover, it would allow them to share more information with the medical professional regarding the pre-stroke life of the survivor required for them to make informed decisions based on the survivor’s preferences for care.” Suggest revising to clarify meaning. This is a complex sentence and may be better as 2 sentences.

On the top of page 23, can you specify the types of emotions? “lack of communication with healthcare professionals has led to improper treatment for…” – can you explain what is meant by this? How did lack of communication lead to “improper” treatment?

On page 38, The quotation isn't simply about going to church but about faith. Is it not? Clarify

In the strengths and limitations can you speak to the benefits and drawbacks of using the 3 data collection approaches to answer this research question?

Reviewer #2: Dear authors,

Thank you for your contributions to such a relevant topic.

Following the acceptance criteria suggested by the PLOS ONE, I consider that:

Overall, the paper is written clearly, with coherent arguments throughout the text, exploring the needs of informal caregivers of stroke survivors. Nevertheless, I believe that across the manuscript, some alterations can be made to improve the consistency and understanding of the manuscript.

I think it is important to clarify the added value of this work since there is already a lot of research on the needs of informal caregivers. In my point of view, the use of the “Grounded Theory Approach” is a differentiating factor but is not presented as such.

Additionally, it is often confused whether this work focuses on the recovery process, or on mid or long-term care. It was reported distinct ideas, such as, “care trajectory”, “recovery”, and “during the transition into the caregiver role”. Therefore, I think that it is central to homogenize the text and make it clear what you focus on.

Furthermore, I believe that the ethical issues underlying this work can be further explored and reflected upon. Despite mentioning that the project was ethically approved and that some measures were taken to protect anonymity, the authors fall short of the central ethical discussions in the development of research on social networks and in studies carried out over the internet

The results presented are dense and presented in detail. However, I feel that the discussion of the main ideas falls short. In particular, the authors described the lack of preparation and uncertainty and then do not explore the educational and formative dimensions of care, the need to invest in health literacy, the need of prepare health professional, the need of interdisciplinary teams that consider the survivors, the informal care and other relatives/family members, etc.

I also recommend being especially careful with "buzz words" namely in the conclusion section. The conclusions must be properly articulated with the presented data and with future steps, inspired by the main findings.

In short, I think that potential readers of this journal will benefit from this work, which can be improved with major revisions.

The specific comments for each section follow below.

Best regards.

Abstract

- I suggest changing the last sentence of the "background" to improve the understanding of the text (e.g., “to identify the needs of caregivers to better support them in their caregiving journey and improve the quality of care delivered.”)

- The results section may not be clear. You choose to put "i.e.,", but you present what appears to be four categories, namely, information, involvement, self-care, and support, am I correct? In the end, do you describe one of them (support dimension) in greater depth, for some reason?

- During the abstract, you imply that the study will focus on the care trajectory, however, in the conclusions, you only present suggestions for the recovery period, which I understand as a short/medium period after hospital discharge. Could you please improve the consistency of the arguments throughout the abstract?

Introduction

- The sentence on line 32 may be excessive, especially if not substantiated with a theoretical reference. The literature has highlighted the complexity of the recovery processes and the influence of multiple factors. I understand and agree about the centrality of care in the recovery of stroke survivors, however, in the way it is exposed, I believe it is excessive.

- Do you really mean “need for stroke caregivers” or do you mean “needs of stroke caregivers”? (lines 41-42)

- It's unclear to me what you mean by "improved understanding". Will it be in-depth? Will it be through other methodologies? Will it be studies that leave clear clues for intervention? To my knowledge, there is a lot of systematic knowledge that understands in-depth (through qualitative methodologies, for example) the needs of survivors. With this sentence (lines 45-47), I cannot understand what is effectively a gap in the literature that you will answer. Additionally, by formulating the last paragraph of the introduction in this way, it seems that you are going to present a systematic review. I believe that it should be improved to immediately understand what is the gap in the literature and what your study brings that is innovative and relevant to this field.

Materials and Methods

- Since the authors intend to publish this work in a journal with a target audience mostly from medicine and health sciences, I suggest the mobilization of fundamental and founders’ authors of the "grounded theory". Furthermore, the definition that you present seems to me to be too general and does not show the added value of using a methodology like this.

- Since you haven't described the data collection procedures yet, I suggest starting the sentence with "A three-phase study was conducted from September 2020 to May 2021” (line 60).

- I don’t believe that “another author (AF) is the right formulation since is the first time that you refer to an author (line 62).

- I suggest “participants” rather than “participant” (line 65).

- Why do you mean by “personal information”? I verified that the sociodemographic characteristics of the participants are presented below. Therefore, I believe that personal information may not have been excluded from the study. Perhaps what you want to explain is that all data identifying the participants were excluded, to ensure anonymity.

- Given the nature of the study, I felt the need for a chapter/paragraph that would reflect on the ethical issues that emerge in this work, which, in my view, are quite a few (especially considering the second and third phases of the study).

- Are you sure that backward citation (lines 75-76) is used to “ensure comprehensiveness”?

- I cannot understand what search expression was used. Were individual searches performed for each expression presented?

- The use of the word “systematic” in this text seems risky to me. To be systematic there are very specific and strict rules. What you present is just a literature review without systematic procedures and results.

- It is necessary for the reader to understand (here or in the results) how many articles were initially retrieved, how many were excluded, which one was included in the “backward citation”, and which were included in the final sample. This becomes central especially since you stated that the review was designed to determine the groups that will be involved in the study. I believe that one flowchart will guide and help the readers, as well as improve the consistency, the quality, and the intelligibility of the study.

- For the quality of the work, it is essential that the methods report consistently and in detail all the steps. In the second phase of the study, it is important to understand which social media platforms were used. What do you understand about popular social media? Cultural differences, for example, change the concept of “popular social media”.

- What do you mean by “systematic search” on social media analysis? Can you please report some bibliographic references in this section?

- Since you are writing in English you should use “dots” instead of “commas” in the numeric formulations (e.g., line 110).

- I believe that the description of the methodology is not done in the best way to be completely clear to the readers. Specifically, in the first paragraph of phase 2 (lines 95-96) you stated that during the analysis you choose how to collect the data inductively. Later, in lines 110 to 111, you stated that the analysis of the post was made deductively according to previous results. Maybe it's wise to put everything together in one sentence or explain it more clearly. It took me several readings to understand clearly what they meant.

- The line in the middle of table 2 suggests that the categories on the right are in opposition to those on the left, perhaps it would eliminate this line.

- You are very vague in the description of phase 3. What kind of questions are they? Closed? Open-ended?

- Theoretically and methodologically, I cannot understand why you choose those age ranges.

Results

- Once again, due to the expression “stroke disease trajectory” (line 157), I cannot understand if your work is about stroke recovery, short, medium, or long-term care.

- Is there any reason to just use two dots instead of an ellipsis? (lines 166-167).

- As a reader, I think that is important to understand the origin of the excerpts presented, at least inform if they come from posts or from the online survey.

- I missed a participants/sample description, even a short one. Having a table without additional information may not be relevant.

- Please, pay attention to the formatting between lines 241-246.

- The results presented are quite dense and I believe it was quite difficult to synthesize them. However, it would be interesting for readers to have a simpler and clearer presentation of the results. Do you think it is possible to present a table with the main results? The categories and some brief explanations?

Discussion

- With results so dense and so complex, I feel that the discussion is very brief. The role of health professionals, health literacy, and the need to diversify available educational materials could be discussed. With so much focus on the lack of preparation, I feel the need to discuss the role of Education Sciences, the preparation of professionals, the diversification of communication channels, etc.

- I don't believe that the authors present a sufficient “strengths and limitations” section. I agree that “face-to-face” interactions could have boosted the results and could be more sensitive and in-depth. However, for example, I think that limitations regarding the use of an online survey in an aging population, with greater technological difficulties are a limitation that should be discussed (sampling bias?).

- In the “implications for future research” section the authors stated, “One plausible approach is the use of technology”. It looks like a statement disconnected from the results and discussion presented. It seems to me that this work has many more implications for future research, in several areas of knowledge.

Conclusion

- I would like to read a conclusion more related to the presented data and not with so many "buzzwords".

6. PLOS authors have the option to publish the peer review history of their article (what does this mean?). If published, this will include your full peer review and any attached files.

Reviewer #1: No

Reviewer #2: No

---

## [Author Response · Author response to Decision Letter 0]

3 Dec 2022

Thank you for your feedback. We have made the changes to the manuscript. Our response to the comments are attached with this revision.

---

## [Editor Report · Decision Letter 1]

18 Jan 2023

Information, Involvement, Self-care and Support - The Needs of Caregivers of People with Stroke: A Grounded Theory Approach

PONE-D-21-28428R1

Dear Dr. Lobo,

We’re pleased to inform you that your manuscript has been judged scientifically suitable for publication and will be formally accepted for publication once it meets all outstanding technical requirements.

Kind regards,

Jeffrey Jutai

Academic Editor

PLOS ONE

Additional Editor Comments (optional):

The authors have satisfactorily addressed the reviewers' concerns.
---

## [Editor Report · Acceptance letter]

23 Jan 2023

PONE-D-21-28428R1 

Information, Involvement, Self-care and Support - The Needs of Caregivers of People with Stroke: A Grounded Theory Approach 

Dear Dr. Lobo:

I'm pleased to inform you that your manuscript has been deemed suitable for publication in PLOS ONE. Congratulations! Your manuscript is now with our production department. 

Kind regards, 

on behalf of

Dr. Jeffrey Jutai 

Academic Editor

PLOS ONE